# Effects of different types and frequencies of early rehabilitation on ventilator weaning among patients in intensive care units: A systematic review and meta-analysis

**Ruo-Yan Wu[1,2], Huan-Jui Yeh[3,4], Kai-Jie Chang[1], Mei-Wun Tsai[2]***

**1** Division of Physical Medicine and Rehabilitation, Taoyuan General Hospital, Ministry of Health and Welfare, Taoyuan, Taiwan, **2** The Department of Physical Therapy and Assistive Technology, National Yang Ming Chiao Tung University, Taipei, Taiwan, **3** Institute of Public Health, National Yang Ming Chiao Tung University, Taipei, Taiwan, **4** The Department of Physical Medicine and Rehabilitation, Taipei Hospital, Ministry of Health and Welfare, New Taipei City, Taiwan

* tmwk@nycu.edu.tw

## Abstract

### Objective

This study aimed to investigate the effects of different types and frequencies of physiotherapy on ventilator weaning among patients in the intensive care unit (ICU) and to identify the optimal type and frequency of intervention.

### Data sources

PubMed, Cochrane Library, EMBASE, and Airiti Library.

### Study selection

Randomized controlled trials that provided information on the dosage of ICU rehabilitation and the parameters related to ventilator weaning were included.

### Data extraction and management

Treatment types were classified into conventional physical therapy, exercise-based physical therapy, neuromuscular electrical stimulation (NEMS), progressive mobility, and multi-component. The frequencies were divided into high ($\geq$ 2 sessions/day or NEMS of > 60 minutes/day), moderate (one session/day, 3–7 days/week or NEMS of 30–60 minutes/day), and low (one session/day, < 3 days/week, or NEMS of < 30 minutes/day).

### Data synthesis

Twenty-four articles were included for systematic review and 15 out of 24 articles were analyzed in the meta-analysis. Early rehabilitation, especially the progressive mobility treatment exerted an optimal effect in reducing the ventilator duration in patients in the ICU (standardized mean difference [SMD] = 0.91; 95% confidence interval [CI] = 0.23–1.58; P < 0.01).

**Funding:** Huan-Jui Yeh is the person received fund.The research was supported by Ministry of Science and Technology of Taiwan (grant no. MOST 108-2314-B-087-001-MY2). The funders played no role in the study design, data collection and analysis, decision to publish, or preparation of the manuscript.

**Competing interests:** The authors have declared that no competing interests exist.

Regarding the treatment frequency, the high-frequency intervention did not result in a favorable effect on ventilator duration compared with the moderate frequency of treatment (SMD = 0.75; 95% CI = -1.13–2.64; P = 0.43).

## Conclusion

Early rehabilitation with progressive mobility is highly recommended to decrease the ventilation duration received by patients in the ICU. Depending on clinical resources and the tolerance of patients, the frequency of interventions should reach moderate-to-high frequency, that is, at least one session per day and 3 days a week.

## Trial registration

**Registration number:** PROSPERO (CRD42021243331).

## Introduction

Patients receiving mechanical ventilation (MV) in the intensive care unit (ICU) are reported to have high in-hospital mortality ranging from 23% to 51%, which was associated with successful weaning and nosocomial infection [1–8]. Prolonged MV may increase the occurrence of complications among critically ill patients [9–12]. Moreover, it could even extend the duration of immobility, which increases the risk of ventilator-associated pneumonia and further restricts patients to MV [13, 14]. Early mobility is believed to improve MV-related outcomes, including ventilator duration and ventilator-free days [15–18]. Early mobility may break the vicious cycle of prolonged MV and immobilization by enhancing the demand of the cardiopulmonary system to avoid respiratory muscle weakness. Furthermore, early rehabilitation was reportedly related to decreased morbidity and mortality [19], disease complications [17, 20], duration of ICU stay, duration of hospital stay [21–24], and rehospitalization rate [25]. Therefore, weaning from MV without delay, through early rehabilitation is important to improve the prognosis of patients with MV in the ICU.

Although increasing evidence highlights that early rehabilitation is helpful for critically ill patients, the intervention program and interest in the outcome are diverse among studies [18, 26]. In this regard, clinicians must consider whether different programs have the same therapeutic effect and must ensure that the prescribed rehabilitation has benefits consistent with previous literature. In addition to the high heterogeneity among early rehabilitation programs, different countries and institutions usually confront two barriers to implementation: a shortage of rehabilitation manpower and a cultural gap in the ICU [27]. In cases of staff shortage and conservative ethos in the ICU, early rehabilitation is usually adjusted to a sustainable level, including the modified programs by in-bed rehabilitation or visiting with lower frequency. Thus, minimal effective dosage of intervention has to be identified for managing patient care and medical manpower.

Ventilator weaning is a goal for clinicians of all disciplines in the ICU; however, the optimal protocols of early rehabilitation for effective ventilator weaning remain unclear. The present systematic review and meta-analysis aimed to investigate the effects of different types and frequencies of early rehabilitation on ventilator weaning of patients in the ICU and to identify the optimal type and frequency of intervention.

## Methods

The present study was conducted in accordance with the Preferred Reporting Items for Systematic Reviews and Meta-Analyses (PRISMA) guidelines and registered on PROSPERO (CRD42021243331; 2021/4/17).

### Study selection

The study selection criteria were based on the Population, Intervention, Comparison, and Outcome (PICO) method. The PICO parameters for this article were as follows: Population, critically ill patients; Intervention, physiotherapy (e.g., active mobilization) with high intensity and high frequency; Comparison, physiotherapy (e.g. passive mobilization) with low intensity and low frequency or control (e.g. medical usual care); Outcome, ventilator weaning. The following inclusion criteria were used for study selection: (1) The target population was the critically ill patients with MV in the ICU rather than in a chronic care center. (2) The interventions had to compare the control programs with lower intensity or frequency with experiment programs with higher dosage. (3) The outcome measures were focused on MV, such as ventilator duration or extubation rate. (4) Randomized controlled trials (RCTs) in English or Chinese published in peer-reviewed journals and the studies provided information on the intervention protocol and dosage. Unpublished manuscripts and conference abstracts were not eligible for study selection. The exclusion criteria were studies without physiotherapy interventions or ventilator-related outcomes, and those focusing on interventions after extubation.

### Data sources and searches

The concatenation of keywords and synonyms by "OR" and "AND" were searched in the following four databases on January 15, 2022: PubMed (1946–2021/12/31), Cochrane Library (1995–2021/12/31), EMBASE (1947–2021/12/31), and Airiti Library (1979–2021/12/31). The keywords included critical illness, intensive care unit, rehabilitation, physical therapy, early mobility, ventilator weaning, and extubation. Every synonym of the keywords was checked with MeSH and the same search protocol was used in each database. The detailed search strategy is shown in S1 Appendix.

Two reviewers (RYW and KJC) independently screened the titles and abstracts of the collected articles. Disagreement was resolved by consensus. Subsequently, a full-text review was conducted. In addition, handsearching was performed on the reference lists of included articles and previously published reviews.

### Quality assessment

Risk of bias 2.0 was used to assess the methodological quality of the recruited articles [28], which was independently scored by two reviewers (RYW and KJC). The assessment items included bias arising from the randomization process, bias due to deviations from intended interventions, bias due to missing outcome data, bias in the measurement of outcome, and bias in the selection of the reported result. If no consensus was reached, a third reviewer (MWT) made the final determination. The inter-reviewer agreement score for quality assessment was calculated as kappa statistics and percentage agreement. If the value of kappa was > 0.75, the inter-reviewer agreement was recognized as "excellent" [29].

### Data extraction

The data and results from the included studies were extracted by using a standardized spreadsheet of Excel (Microsoft Excel 2016; Microsoft Corp., Redmond, WA, USA) that documented

basic information regarding the study (e.g., first author, year of publication, country, email address of the corresponding author); characteristics of participants (e.g., medical conditions, sedation control, mean age, and sample size); type of intervention for the experimental group (e.g., name of the program, components of the program, intervention time, and delivery frequency); type of intervention for the control group (same as above); and ventilator-related outcomes (e.g., extubation rate, re-intubation rate, ventilator duration, ventilator-free days during the first 28 days of hospital stay, weaning duration, intubation duration, and re-ventilation duration). Outcomes reported as continuous variables were presented as means ± standard deviations. If only the median and interquartile range were reported, they were converted to mean and standard deviation using appropriate statistical formulas [30]. The outcomes reported as categorical variables were presented as the event rate. All data were extracted independently by two reviewers (RYW and KJC). When results were missing or not fully reported, efforts were made to contact the contributing authors to retrieve missing data.

## Data management

The types of treatment were grouped into four categories according to the intervention program: (1) Conventional physical therapy (CPT), which included positioning, stretching, range of motion exercise, and chest physical therapy, such as manual hyperinflation, chest expansion exercise, postural drainage, and suction. (2) Exercise-based physical therapy (EPT), which included strengthening, balance training, and cycling. (3) Neuromuscular electrical stimulation (NEMS). (4) Progressive mobility (PM), which progressed the program according to patient ability. A passive range of motion is the first step and is used for patients unable to follow instructions. As the patients become more responsive, the intervention advanced step-by-step from lying to sitting, sitting on the edge of the bed, transferring to a chair, standing, stepping in place, and walking. (5) The multiple treatment elements were combined as the multi-component.

For categorizing treatment frequency, studies were classified as follows: (1) High frequency, with $\geq 2$ sessions per day or NEMS of $\geq 60$ minutes per day. (2) Moderate frequency, with one session per day, 3–7 days per week, or NEMS of 30–60 minutes per day. (3) Low frequency, with one session per day, $< 3$ days per week, or NEMS of $< 30$ minutes per day.

Depending on the program used for the control group, the included articles were categorized as either active control, in which the control group only received medical treatment or routine care with chest care from the ICU team, or active control, in which low-intensity or low-frequency physiotherapy was used as the control protocol.

## Data synthesis and analyses

First, a qualitative systematic review was conducted for all included articles. The narrative and numerical description were used to summarize the characteristics and the result of the studies.

Meta-analysis was performed when the number of articles with the same ventilator-related outcome was more than three. The articles were excluded from pooling when there were missing data, insufficient treatment information, or difficulty in the grouping. The detailed process of article selection for quantitative synthesis is shown in Fig 1. The data synthesis used Review Manager 5.4 (The Cochrane Collaboration, London, UK) with statistical significance set at a P-value of $< 0.05$. Categorical variables were compared using odds ratios (ORs), while continuous variables were compared using standardized mean difference (SMD). Moreover, 95% confidence intervals (CIs) were calculated by Wald-type methods in Review Manager 5.4 for all values. Heterogeneity among articles was assessed using the Q test and $I^2$ statistic. When

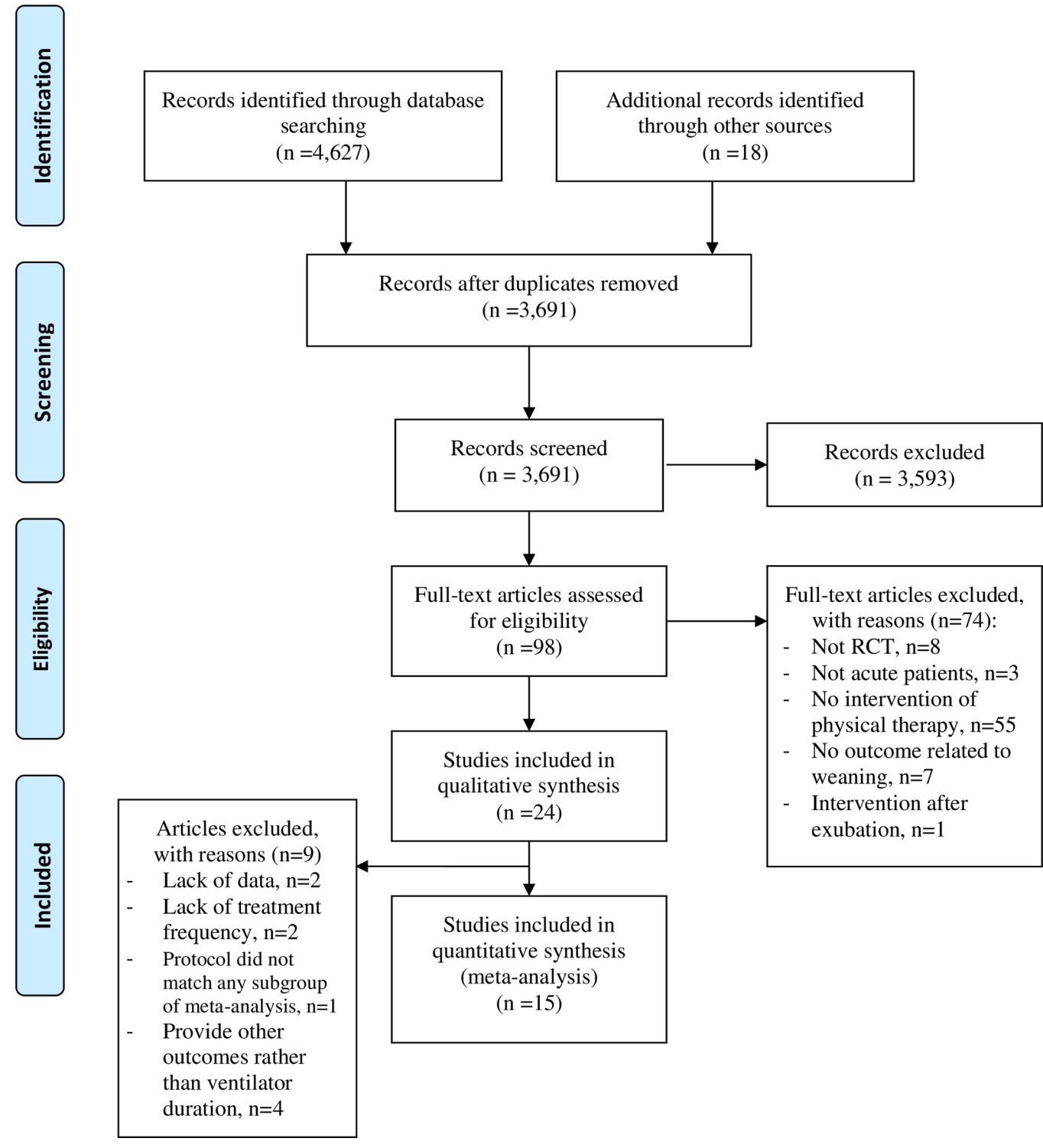

**Fig 1. PRISMA flow diagram.**

the $I^2$ of overall heterogeneity was > 50%, pooling data were analyzed in a random-effects model. Publication bias was evaluated by funnel plot.

## Results

### Study recruitment

A total of 3,673 articles were eligible in the electronic search, and additional 18 articles were included from handsearching. Ninety-eight articles were selected after screening the titles and abstracts. During the full-text review process, 74 articles were excluded (i.e., non-RCTs, articles recruiting chronic patients, lacking physiotherapy intervention, lacking ventilator-related outcomes, and focusing on post-extubation interventions). A total of 24 articles were included for review (Fig 1).

### Risk of bias

Regarding article quality, there were 5, 11, and 8 articles of high, moderate, and low quality, respectively. The scores given for each item and the combined results are shown in Figs 2 and 3. Seventeen articles (70.8%) did not score on the blinding. The inter-reviewer agreement of quality assessment was excellent (kappa = 0.84, percentage agreement = 91.7%).

### Description of 24 included articles

A total of 2,567 individuals were recruited across these 24 articles, and > 50% of these were from Europe or the United States. Most were critically ill patients in the ICU, including medical, surgical, and respiratory ICUs. Multi-component treatment was the most common type of intervention. Combined treatment elements primarily involved PM, with 68.82% of patients receiving this intervention undertaking out-of-bed exercise. Only 3% of patients received NEMS which was the rarest intervention (Table 1).

As summarized in S1 Table, seven (29%) of 24 studies were found to have positive effects on the ventilator duration, ventilator-free days, or extubation rate during hospitalization [31–37]. All 4 studies using the PM approach had significantly shorter ventilator duration in the intervention group; however, differences in the ventilator-free days were not significant [31–33, 35]. One of the studies using early mobilization with/without an elastic band was beneficial with respect to ventilator duration when compared to multiple components including passive and active range of motion and breathing exercises [34]. A study using rotation therapy (changing position continuously for 18 h/day) and percussion showed significantly shorter ventilator duration and longer ventilator-free days than that of routine position changing every 2–4 h [36]. A study using multimodality chest physical therapy showed a higher extubation rate when compared to studies using manual hyperinflation and suctioning [37]. Regarding multiple-component treatments as early rehabilitation intervention, 83% (10/12) of studies did not show significant benefit either in the outcomes related to ventilator weaning, when compared to the low dosage of intervention, medical treatment, or usual care as the control group [38–48]. Three studies using CPT, which were primarily comprised of chest physical therapy and range of motion exercise, did not show significant improvement in weaning outcomes [49–51]. All four studies using NMES alone or combined exercises as an intervention did not show significant benefit on ventilator weaning [41, 42, 52, 53]. Only one study indicated that treatment combining physical therapy and airway clearance techniques may lead to longer ventilator duration than decubitus care and tracheal suctioning [54]. Thirteen (54%) of the 24 included studies provided the conditions of sedation control. The methods of sedation control during intervention included maintaining the sedation level within the Richmond Agitation-Sedation Scale (RASS) range of -1 to 1 [38, 40, 45, 47] or Ramsay sedation scale range of

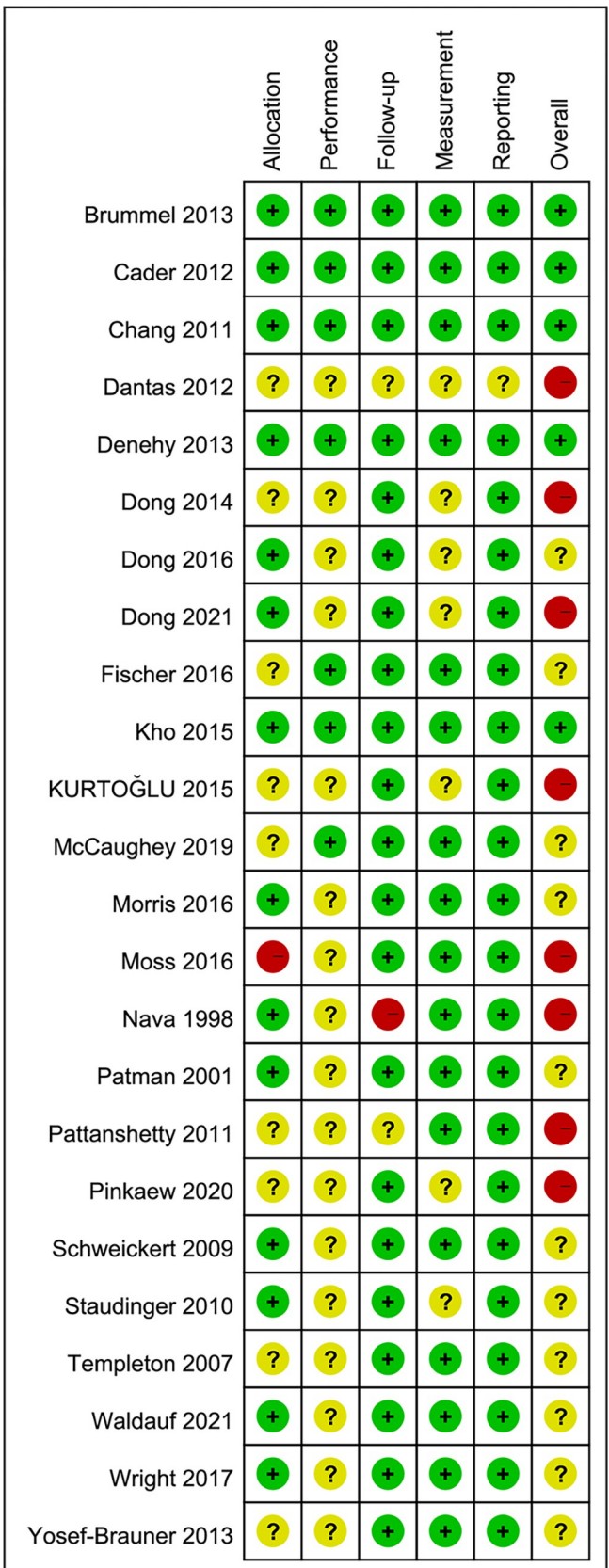

**Fig 2. Risk of bias summary.**

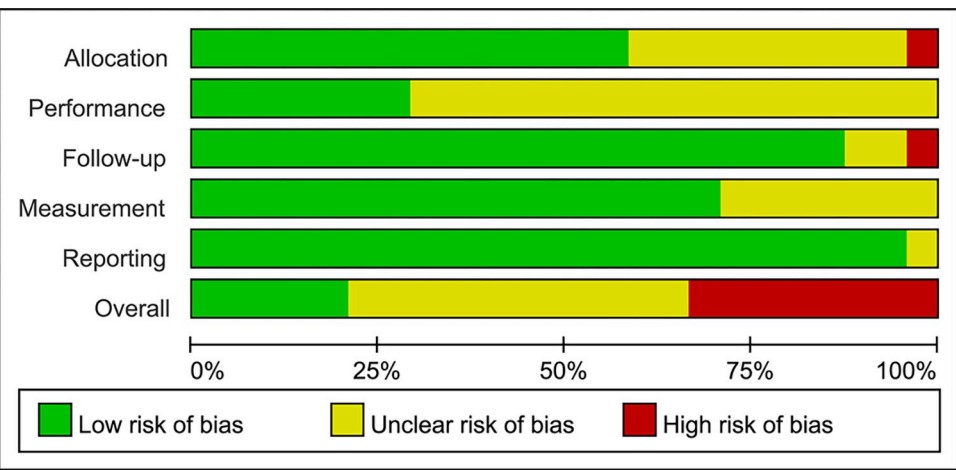

**Fig 3. Risk of bias graph.**

3 to 5 [36], recording the sedation levels [41, 43, 44, 52], and training after sedatives withdrawal or sedation interruption for 2 h before training [31, 32, 35, 39].

## Meta-analysis of 15 included studies

Fifteen articles that used ventilator duration as an outcome measure met the criteria for meta-analysis. Regarding the types of treatment protocols, there were eight articles pooled for the

**Table 1. Demographics of included patients.**

| Demographics | Total n = 2,567 |
|---|---|
| **Geographical region (%)** | Europe: 914 (35.61%) |
| | USA: 645 (25.13%) |
| | Asia: 542 (21.11%) |
| | Australia: 380 (14.80%) |
| | South America: 56 (2.18%) |
| | Turkey: 30 (1.17%) |
| **Setting (%)** | ICU: 1,045 (40.71%) |
| | MICU: 833 (32.45%) |
| | SICU: 609 (23.72%) |
| | RICU: 80 (3.12%) |
| **Participants' type (%)** | Critical illness: 1,955 (76.16%) |
| | Post-cardiac surgery: 370 (14.41%) |
| | Pulmonary: 196 (7.64%) |
| | Bed-ridden elderly: 28 (1.09%) |
| | ICUAW: 18 (0.7%) |
| **Intervention type (%)** | CPT: 767 (29.89%) |
| | EPT: 0 (0%) |
| | NMES: 74 (2.88%) |
| | PM: 350 (13.63%) |
| | Multi-component: 1,376 (53.60%) |

CPT: Conventional physical therapy; EPT: Exercise-based physical therapy; ICU: Intensive Care Unit; ICUAW: Intensive Care Unit-Acquired Weakness; MICU: Medical Intensive Care Unit; NMES: Neuromuscular Electrical Stimulation; PM: Progressive mobility; RICU: Respiratory Intensive Care Unit; SICU: Surgery Intensive Care Unit

meta-analysis, which used medical treatment or usual care as a comparison [31–33, 36, 37, 50–52]. The analysis revealed that early rehabilitation significantly reduced the ventilator duration in patients in the ICU (SMD = 0.39; 95% CI = 0.01–0.78; P = 0.04; heterogeneity test, $\chi^2$ = 51.57, P < 0.001; $I^2$ = 86%). In the subgroup analysis of treatment types, only early rehabilitation with PM had a significant and optimal treatment effect on ventilator duration (SMD = 0.91; 95% CI = 0.23–1.58; P < 0.01) (Fig 4A). However, in contrast to the control protocol with lower-dose early rehabilitation, the experimental protocol with higher-dose early rehabilitation did not significantly reduce ventilator duration (SMD = 0.19; 95% CI = -0.27–0.66; P = 0.41; heterogeneity test, $\chi^2$ = 17.45, P = 0.002; $I^2$ = 77%) (Fig 4B) [34, 40–42, 44].

Regarding the frequencies of the treatment protocol, the meta-analysis using medical treatment or usual care as a comparison was composed of 7 articles [31–33, 36, 37, 50, 52]. The overall effect of early rehabilitation significantly reduced the ventilator duration of patients in the ICU (SMD = 0.47; 95% CI = 0.03–0.91; P = 0.04; heterogeneity test, $\chi^2$ = 43.61, P < 0.001; $I^2$ = 86%) (Fig 5A). In comparison with the moderate frequency of treatment, the high-frequency intervention did not show a significant favorable effect on ventilator duration (SMD = 0.75; 95% CI = -1.13–2.64; heterogeneity test, $\chi^2$ = 10.66, P = 0.001; $I^2$ = 91%) (Fig 5B) [45, 46].

## Discussion

This systematic review and meta-analysis highlighted the benefits of early rehabilitation on ventilator weaning, which could shorten ventilator duration. Furthermore, our findings demonstrated that early rehabilitation with PM protocol could administer the best therapeutic effect, and those programs involving proactive protocol tend to reduce ventilator duration. Regarding treatment frequency, early rehabilitation with different frequencies can decrease ventilator duration compared to medical treatment or usual care. The effect of an early rehabilitation program with the high-frequency intervention was not significantly different from the moderate-frequency program.

Our meta-analysis provided evidence for various ICU rehabilitation programs on ventilator weaning, which was in addition to traditional mid-term or long-term outcome indices, such as physical function, quality of life, and duration of hospitalization. Although MV is lifesaving, ventilator-associated events, including infectious and non-infectious complications, presented longer MV use, longer ICU stays, and even higher mortality [55, 56]. Compared to other functional outcomes, successful weaning from MV or not weaning is a visible early outcome to increase the ICU team's awareness and acceptance of early rehabilitation. Among various measures related to the ventilator, ventilator duration, which is directly related to the risk of ventilator-associated events development, is the most common outcome measure in previous studies. Ventilator-associated complications decreased the probability of weaning from MV by 30% and significantly increased the ventilator duration (17 vs 6.2 days, P < 0.05) [56]. Peña-López et al. suggested that reducing the exposure or the duration of MV should be a priority to limit ventilator-associated complications [57]. These findings highlighted ventilator duration as a sensitive clinical outcome to reflect successful extubation [56–58].

Regarding treatment type, the previous meta-analysis indicated that early mobilization was beneficial for extubation, including higher ventilator-free days and shorter ventilator duration [16, 17]. However, these studies focused on how effective the rehabilitation was in the prevention or reversal of weaning failure and other clinically important outcomes. It remains unknown which intervention is the best among the various protocols and barriers to clinical application still exist due to the high heterogeneity of the intervention. Our findings could provide further information in this regard. Early rehabilitation was roughly categorized into

## A. Physical therapy compared to medical treatment

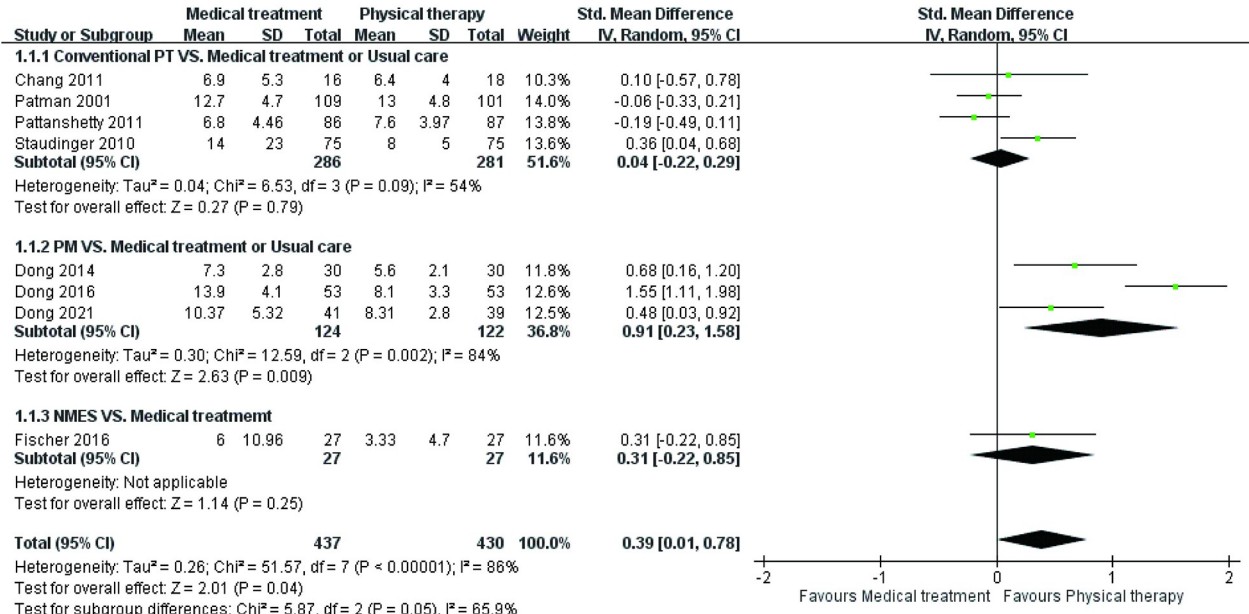

## B. Low-dose physical therapy compared to high-dose physical therapy

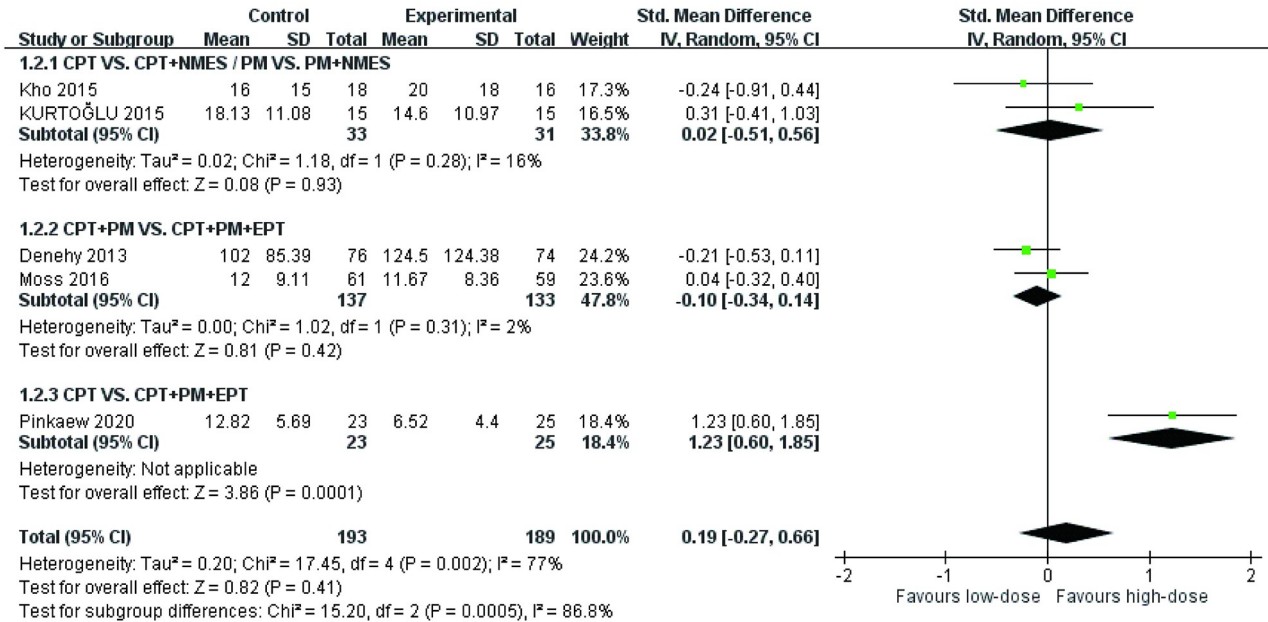

**Fig 4. Forest plot: The effects of different types of physiotherapy on ventilator duration.** (A) Physical therapy compared to medical treatment; (B) low-dose physical therapy compared to high-dose physical therapy.

passive mobilization, active mobilization, and progressive exercise and mobility [59]. Previous studies have found that PM not only decreased the durations of ICU and hospital stay (5.5 vs 6.9 days, P = 0.025, and 11.2 vs 14.5 days, P = 0.006, respectively) [60], but also improved

A. Physical therapy compared to medical treatment

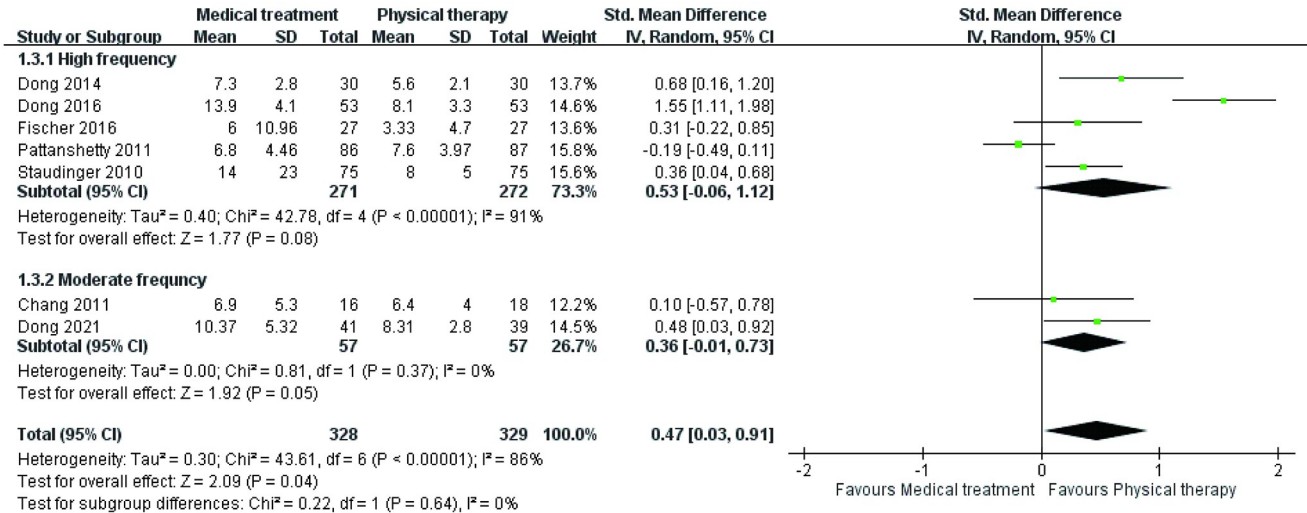

B. Moderate-frequency physical therapy compared to high-frequency physical therapy

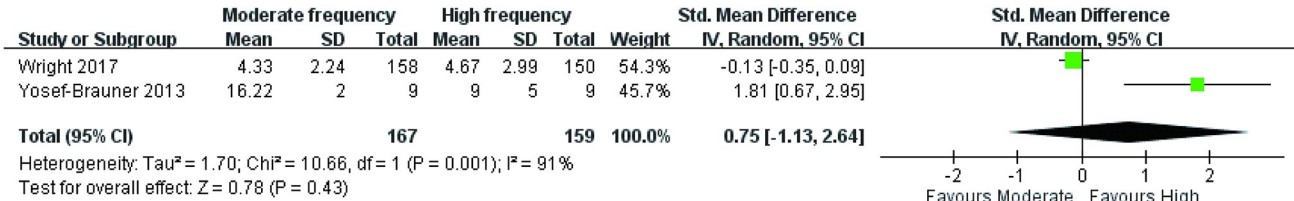

**Fig 5. Forest plot: The effects of different frequencies of physiotherapy on ventilator duration.** (A) Physical therapy compared to medical treatment; (B) moderate-frequency physical therapy compared to high-frequency physical therapy.

functional status at discharge from ICU when compared to those of the conventional treatment without a pre-established routine [60, 61]. They also indicated that an increased activity achieved by PM was associated with increased functional outcomes. The increased activity, such as out-of-bed program, which was part of daily activities, could maintain previous functionality and a return to functional independence [60, 61]. Clini et al. further stated that in patients who experienced difficulty in weaning in the respiratory ICU, an increased gained functional recovery resulted in a higher likelihood of weaning from ventilator [62]. A possible explanation could be that the intensity reached by PM would recruit both musculoskeletal and cardiopulmonary systems simultaneously, which helps in the functional recovery and the successful weaning of MV. Recovery of adequate inspiratory muscle force may be the major determinant of successful weaning, as it allows patients to tolerate the respiratory load and breathe far below the diaphragm fatigue threshold [63]. Besides, the programs of CPT in our meta-analysis compared with medical treatment or usual care were mainly comprised of in-bed programs with chest physical therapy and range of motion exercises, which were relatively gentle [36, 37, 51]. Only one study applied physical therapy in the sitting position [50]. Insufficient dosage may be the reason for the non-significant effect of treatment on ventilator duration. Regarding the application of NMES, the limited effects on ventilator weaning were predictable as the investigators focused on the limb muscles rather than the respiratory muscles [41, 42, 52, 53].

As increasing evidence confirms the benefits of early rehabilitation, it is unethical to design a control group without intervention. Various studies have used multi-component programs with relatively low intensity as a comparison to identify the effect of the specific component. According to our meta-analysis, the overall effect of intervention programs was not superior to controls [34, 40–42, 44]. However, the non-significant result does not mean that these protocols did not have positive effects on ventilator duration. We found a low magnitude of difference in the amount of activity level between the groups in the studies with multi-component. When the exercise intensity between groups was too similar, it was difficult to identify the additional effect of the intervention protocol. This inference could be supported by the results of subgroup analysis in Fig 4B. When the EPT was the only extra component in the intervention group, the intervention protocol was not significantly better than the control program [40, 44]. As the gap of intervention intensity enlarged, the additional components consisting of EPT and PM had a significant benefit on ventilator duration [34]. On the other hand, most EPTs in our included studies were limb-strengthening exercises and had limited benefit on ventilator weaning. By contrast, inspiratory muscle strengthening may be a choice when the ICU team focused on the respiratory system [49].

Regarding treatment frequency, there is a lack of consensus since the protocol would change with disease category. For instance, patients with neurological disorders in the ICU had a lower frequency of early rehabilitation (median = 2.1 sessions per week) [64]. Although our meta-analysis showed early rehabilitation trended to reduce ventilator duration regardless of the intervention frequencies, the protocol with low frequency ($< 3$ days per week or NEMS of $< 30$ minutes per day) was rarely used in our included studies and did not have a significant effect [42]. In the clinic, the ICU team needs to constantly consider whether the treatment dose is sufficient or not. Contrary to out-of-bed exercise, in-bed exercise like chest physical therapy was easier for increasing the frequency, regardless of the demand of the patient's physical fitness or the cost of treatment. Therefore, calculating the total duration of treatment or considering the type and frequency of treatment simultaneously may be appropriate to quantify the mobilization dose. A cohort study in 2021 implemented this concept. They used the mobilization quantification score (MQS), which combined the activity level and duration, and indicated that a high dose of mobilization is an independent predictor for functional ability after patients are discharged from hospitals [65]. However, the relevant data are too scarce to synthesize from the present studies. We suggested that a scale to quantify early mobilization, such as the MQS, should be used in future research.

In addition, a higher treatment dose did not necessarily mean that it was more effective. A recent clinical trial, in which both intervention and usual care groups had a high frequency of early mobilization, indicated that an increase in early mobilization was not associated with a longer survival time. Furthermore, greater exposure to mobilization in the intervention group led to increased adverse events, such as oxygen desaturation [66]. The findings were similar to those of our meta-analysis that compared moderate-frequency treatment with high-frequency intervention. The excessive intensity of treatment may exceed the tolerance of patients and become a burden on patients in the ICU. Clinical staff should consider the optimal dose of early rehabilitation for ventilator weaning rather than increase the treatment frequency endlessly.

In summary, a sufficient dose of mobilization consisted by type and frequency is the key factor to decrease ventilator use. For clinical applications, PM is the type recommended. According to the progression procedures in our included articles, RASS would assist physical therapists to decide the initial level applied to patients in the ICU [38]. Passive range of motion was for coma patients (RASS -4 to -5) and those who were aroused to voice (RASS -3 to -2) could try sitting. Alert patients (RASS -1 to 1) could have the most options, such as active

exercise, sitting at the edge of the bed, standing, and walking. As the description mentioned in the Method, the progressive tasks were similar between articles, which focused on functional mobility. However, the progression milestones varied among different studies [31, 32, 35, 40, 41, 44, 46, 47]. Irrespective of the milestones chosen for implementation, clinical staff should set the stopping criteria for the intervention and must follow the patient's tolerance. However, when clinical staff cannot handle out-of-bed exercises, an in-bed program with increased frequency according to patient tolerance is an alternative to reach a higher dose. As a single session of chest physiotherapy is less labor-intensive and less time-consuming, increasing the frequency of daily treatment is also a contingency to achieve more benefits.

There were several limitations to this study: (1) there were 24 studies that could be included in the systematic review; however, only 15 studies were eligible to be included in the meta-analysis. After stratifying with type or frequency of intervention in comparison to medical treatment or low frequency, there were only fewer than five articles included in each meta-analysis. (2) Among the included articles with a low-to-moderate quality, the major risk of bias was a lack of blinding and placebo or true control. Thus, more high-quality RCTs with larger sample sizes in this field are essential. (3) Considering the diversity of ventilation-related outcome indicators and the small number of included studies, only the studies using the ventilation duration were included in the meta-analysis to ensure the reliability of the study and comparison. (4) Analyses of moderator variables on the effects of early rehabilitation programs (e.g, age, sex, sample size, comorbidities, and sedation level) were not performed. Lesser clinical prognostic factors could be identified for the treatment plan. (5) Although the sedation level of individuals may affect their active participation in the intervention, only half of the included studies provided individuals' conditions of sedation with various sedation controls among studies. Although we used the random-effects model for pooling, the high heterogeneity of sedation control might not be neglected and should be considered in clinical practice. (6) During the meta-analysis, there was one study that provided the median rather than the mean data and we did not receive a response from the authors after contact via email [52]. In this study, we estimated the means based on formulas from the literature; however, some data extraction bias remains inevitable. (7) Publication bias was not suitable to be evaluated by funnel plot and the impact of bias may exist.

Finally, the multi-component intervention program, usually designed by combinations of any two to three interventional elements, is also flexible, variable, and easily applied in the clinical setting. No significance was found when compared to alternative active control in this review. There is uncertainty regarding combining intervention elements for effectiveness. In the future, with the increase of research on the different protocols of early rehabilitation for patients in ICU, we plan to update this systematic review and meta-analysis. A network meta-analysis may be attempted to explore the association between the complexity and variety of multi-component protocols and outcomes.

## Conclusions

In terms of ventilator withdrawal, different types have different benefits in patients in the ICU. Programs with progressive mobility are the most recommended prescription. Moreover, depending on clinical resources and the tolerance of patients, the frequency of interventions should reach moderate-to-high frequency that treats at least one time a day, 3 days a week. The study results may serve as an empirical basis for devising intervention plans when there is insufficient medical manpower to develop early rehabilitation in clinical units.

## Supporting information

**S1 Checklist. PRISMA checklist of systematic review.**
(DOC)

**S1 Appendix. Search strategy.**
(DOCX)

**S1 Table. Characteristics of recruited studies.**
(DOCX)

## Acknowledgments

We appreciate Cardiopulmonary Rehabilitation and Health Promotion Lab of National Yang Ming Chiao Tung University for their encouragements and suggestions.

## Author Contributions

**Conceptualization:** Ruo-Yan Wu, Huan-Jui Yeh, Mei-Wun Tsai.

**Data curation:** Ruo-Yan Wu, Kai-Jie Chang, Mei-Wun Tsai.

**Formal analysis:** Ruo-Yan Wu, Kai-Jie Chang.

**Funding acquisition:** Huan-Jui Yeh.

**Investigation:** Ruo-Yan Wu, Huan-Jui Yeh, Kai-Jie Chang.

**Methodology:** Ruo-Yan Wu, Kai-Jie Chang.

**Project administration:** Huan-Jui Yeh, Mei-Wun Tsai.

**Supervision:** Huan-Jui Yeh, Mei-Wun Tsai.

**Visualization:** Ruo-Yan Wu.

**Writing – original draft:** Ruo-Yan Wu.

**Writing – review & editing:** Huan-Jui Yeh, Mei-Wun Tsai.

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
