## [Decision Letter · Decision Letter 0]

20 Dec 2022

PONE-D-22-29326Effects of different types and frequencies of early rehabilitation on ventilator weaning among patients in intensive care units: A systematic review and meta-analysisPLOS ONE

Dear Dr. Tsai,

Thank you for submitting your manuscript to PLOS ONE. After careful consideration, we feel that it has merit but does not fully meet PLOS ONE’s publication criteria as it currently stands. Therefore, we invite you to submit a revised version of the manuscript that addresses the points raised during the review process.

We look forward to receiving your revised manuscript.

Kind regards,

Steven E. Wolf, MD

Academic Editor

PLOS ONE

Journal Requirements:

Additional Editor Comments:

Editor - Thank you for submitting your paper to us for review. I sent it to six distinguished referees for comment and decision of whom two agreed to review; you will see these below. They thought that the paper has merit, but each have raised some substantial issues to be addressed in a revision. Please carefully consider the comments below and reply directly to each in a cover letter with appropriate marked and linked changes to the manuscript. I look forward to seeing the revision, which I will send back to the same referees for further comment and decision. Please understand that this is not a guarantee of future publication, as the revised manuscript itself must stand on its own merit.

Reviewers' comments:

Reviewer's Responses to Questions

**Comments to the Author**

1. Is the manuscript technically sound, and do the data support the conclusions?

Reviewer #1: Partly

Reviewer #2: Yes

2. Has the statistical analysis been performed appropriately and rigorously? 

Reviewer #1: Yes

Reviewer #2: I Don't Know

3. Have the authors made all data underlying the findings in their manuscript fully available?

Reviewer #1: Yes

Reviewer #2: No

4. Is the manuscript presented in an intelligible fashion and written in standard English?

Reviewer #1: Yes

Reviewer #2: No

5. Review Comments to the Author

Reviewer #1: My pleasure to review the study: "Effects of different types and frequencies of early rehabilitation on ventilator weaning among patients in intensive care units: A systematic review and meta-analysis." The goal of the study was to investigate the effects of different types and frequencies of physiotherapy on ventilator weaning among patients in the ICU, and to identify the optimal type and frequency of the intervention. The conclusion is that early rehabilitation with progressive mobility is recommended to decrease ventilator duration.

Some concerns:

Introduction:

I suggest reducing the Introduction by 25%. The authors must be discuss only the goal of the study.

Methods: OK

Results:

How will the authors control sedation levels?

Discussion:

I would like to read the opinion of the authors about their findings (the step increase of physical activity). This is a important finding and must be better discussed.

And, how the explanation regarding no effectiveness of higher activity. Because this is a trend in the current medical literature.

References:

Some authors are abbreviated inadequately.

Figures:

The figures do not have good quality.

Reviewer #2: I congratulate the authors for such work. It provides an important contribution to the field of rehabilitation.

Please consider consulting with a person well-versed in English grammar.

INTRODUCTION

Please consider how the examined interventions might help in weaning from ventilator.

METHODS

Please specify whether reports such as unpublished manuscripts or conference abstracts were eligible for inclusion

Consider providing rationale for any notable restrictions to study eligibility (e.g., articles focusing on patients with chronic conditions.

How many and what were the Chinese languagues?

Please consider specifying the date when each source was last searched or consulted

Specify restrictions on dates

Specify a bit more what “other sources” shown in S1 Appendix are

Specify how articles were chosen from hand-searching references (what criteria were applied)?

Provide more details on search strategy as it was performed in each database.

Did you need to use special tools to convert search terms between English and Chinese?

How were the articles translated?

How was the data extracted (what software)?

Under Data Synthesis/Analysis, were there any limitations set to interpret the size of effect?

What meta-analysis method was used?

Describe in detail process used to decide which studies of the 24 were eligible for synthesis (15?).

Please consider specifying method used to calculate the confidence interval for the summary effect (e.g., Wald-type CI, Hartung…..).

RESULTS

Please consider mentioning some studies that might have met inclusion criteria but were excluded from analyses and why?

Please consider not starting consecutive sentences with “One study….”

DISCUSSION

Please consider discussing limitations of your review process if any, and discuss impact of each limitation. You already discussed limitation of the evidence.

Please consider providing recommendations for future research to address short comings of this study.

6. PLOS authors have the option to publish the peer review history of their article (what does this mean?). If published, this will include your full peer review and any attached files.

Reviewer #1: **Yes: **Cassiano Teixeira

Reviewer #2: No

---

## [Author Response · Author response to Decision Letter 0]

16 Feb 2023

Journal Requirements:

Reply: 

We confirmed that the revised manuscript met all the requirements of PLOS ONE.

Reply:

We confirmed that all data underlying the findings are fully available without restriction. All relevant data were within the revised manuscript and its supporting materials. We provided the information in the revised cover letter.

Additional Editor Comments:

Editor - Thank you for submitting your paper to us for review. I sent it to six distinguished referees for comment and decision of whom two agreed to review; you will see these below. They thought that the paper has merit, but each have raised some substantial issues to be addressed in a revision. Please carefully consider the comments below and reply directly to each in a cover letter with appropriate marked and linked changes to the manuscript. I look forward to seeing the revision, which I will send back to the same referees for further comment and decision. Please understand that this is not a guarantee of future publication, as the revised manuscript itself must stand on its own merit.

Reply:

We revised the manuscript carefully to address all questions, suggestions, and insights. Please see the responses for each comment as below.

Reviewer Comments:

Reviewer #1: 

Introduction

1. I suggest reducing the Introduction by 25%. The authors must be discuss only the goal of the study.

Reply:

Thanks for your comment. In order to focus on the early rehabilitation and ventilator weaning, the Introduction was rewrote (Page 4-5, Lines 47-78). We finally reduced the Introduction by 24.1% (from 473 to 359 words). 

Results

2. How will the authors control sedation levels?

Reply:

There were only 13 of 24 included studies described the conditions of sedation control. According to the method of sedation control, the included articles could be categorized into 4 conditions. We added the conditions of sedation control for each included study into the Supplementary material (S2 Table). The methods of sedation control were summarized in the Results as below.

“Thirteen (54%) of the 24 included studies provided the conditions of sedation control. The methods of sedation control during intervention included maintaining the sedation level within the Richmond Agitation-Sedation Scale (RASS) range of -1 to 1 [38, 40, 45, 47] or Ramsay sedation scale range of 3 to 5 [36], recording sedation levels [41, 43, 44, 52], and training after sedatives withdrawal or sedation interruption for 2 hours before training [31, 32, 35, 39].” (Page 14, Lines 227-233)

We also added the limitation related to reports of original studies about sedation condition in the Discussion as below. 

“(5) Although the sedation level of subjects may affect their active participation in the intervention, only half of the included studies provided subjects’ conditions of sedation with various sedation controls among studies. Although we used the random-effects model for pooling, the high heterogeneity of sedation control might not be neglected and should be considered in clinical practice.” (Page 23, Lines 398-402) 

Discussion

3. I would like to read the opinion of the authors about their findings (the step increase of physical activity). This is an important finding and must be better discussed. And, how the explanation regarding no effectiveness of higher activity. Because this is a trend in the current medical literature.

Reply:

Regarding the step increase of physical activity, the progressive tasks were similar between articles, which focused on functional mobility. Passive range of motion is the first step and is used for patients unable to follow the instructions. As the patients become more interactive, the intervention moves patients as soon as possible from lying to sitting, sitting on edge of bed, transferring to chair, standing, stepping, and walking. However, the progression milestones were various in different studies. For example, Dong et al. advanced the program to the next level when patients could reach the target, such as sitting for 20 minutes and standing for 5 minutes. The progression conducted by Yosef-Brauner et al. depended on the ability of the extremities to resist gravity. Moss et al. fixed a 25% increase for ambulation distance when patients could bear. Denehy et al. kept Borg score between 3 to 5 for every treatment level. No matter what milestones the researchers used, all articles would set the criteria for stopping the intervention and follow the patients’ tolerance.

As advised in the comment, we provided more information for clinical implementation and the manuscript was revised as 

“According to the progression procedures in our included articles, RASS would assist physical therapists to decide the initial level applied to ICU patients [38]. Passive range of motion was for coma patients (RASS -4 ~-5) and those who were aroused to voice (RASS -3~-2) could try sitting. Alert patients (RASS -1~1) could have the most options, such as active exercise, sitting at the edge of the bed, standing, and walking. As the description in the method, the progressive tasks were similar between articles, which focused on functional mobility. But the progression milestones varied with different studies [31, 32, 35, 40, 41, 44, 46, 47]. Irrespective of the milestones chosen for implementation, clinical staff should set the stopping criteria for the intervention and must follow the patient’s tolerance.” (Page 21-22, Lines 370-379)

And we added a paragraph for the explanation regarding no effectiveness of higher activity. The manuscript was revised as 

“In addition, a higher treatment dose did not necessarily mean that it was more effective. A recent clinical trial in which both intervention and usual care groups had a high frequency of early mobilization indicated that an increase in early mobilization was not associated with a longer survival time. Furthermore, greater exposure to mobilization in the intervention group led to increased adverse events, such as oxygen desaturation [66]. The findings were similar to our meta-analysis that compared moderate-frequency treatment with high-frequency intervention. The excessive intensity of treatment may exceed the tolerance of patients and become a burden on ICU patients. Clinical staff should consider the optimal dose of early rehabilitation for ventilator weaning rather than increase the treatment frequency endlessly.” (Page 20-21, Lines 358-367)

References

4. Some authors are abbreviated inadequately.

Reply:

Thanks for your feedback. All abbreviation of authors and pages of starting and ending in the References were corrected and confirmed. (remarking with highlight, Page 30-41 in revised manuscript with track changes; Page 25-36 in manuscript) 

Figures

5. The figures do not have good quality.

Reply:

Thanks for your feedback. The pixels of figures were adjusted to 600dpi. Please see revised figures. 

Reviewer #2: 

1. Please consider consulting with a person well-versed in English grammar.

Reply: 

Thanks for your feedback. The manuscript was proofread and edited by native speaker. Most of the changes were to improve the readability or logical flow of the document. The English editing certificate is attached in the file (Response to Reviewers).

Introduction

2. Please consider how the examined interventions might help in weaning from ventilator.

Reply:

Thanks for your feedback. After considering the comments from you and the other reviewer who suggested to reduce the Introduction by 25%, we revised Introduction as “Early mobility is believed to improve MV-related outcomes, including ventilator duration and ventilator-free days [15-18]. Early mobility may break the vicious cycle of prolonged MV and immobilization by enhancing the demand of the cardiopulmonary system to avoid respiratory muscle weakness.” (Page 4, Lines 53-57)

And we discussed possible mechanism of interventions in the Discussion. Here is the paragraph about how the examined interventions might help in ventilator weaning.

“Early rehabilitation was roughly categorized into passive mobilization, active mobilization, and progressive exercise and mobility [59]. Previous studies found that PM not only decreased the durations of ICU and hospital stay (5.5 vs 6.9 days, P = 0.025, and 11.2 vs 14.5 days, P = 0.006, respectively) [60], but also improved functional status at discharge from ICU when compared to those of the conventional treatment without a pre-established routine [60, 61]. They also indicated an increased activity achieved by PM was associated with increased functional outcomes. The increased activity, such as out-of-bed program which was part of daily activities, could maintain previous functionality and a return to functional independence [60, 61]. Clini et al. further stated that in patients who experienced difficulty in weaning in the respiratory ICU, the more functional recovery they gained, the more likelihood of weaning from ventilator [62]. A possible explanation could be that the intensity reached by PM would recruit both musculoskeletal and cardiopulmonary systems simultaneously, which helps in the functional recovery and the successful weaning of MV. Recovery of adequate inspiratory muscle force may be the major determinant of successful weaning since it allows patients to tolerate the respiratory load and breathe far below the diaphragm fatigue threshold [63]. Besides, the programs of CPT in our meta-analysis compared with medical treatment or usual care were mainly comprised of in-bed programs with chest physical therapy and range of motion exercises, which were relatively gentle [36, 37, 51]. Only one study applied physical therapy in the sitting position [50]. Insufficient dosage may be the reason for the nonsignificant effect of treatment on ventilator duration. Regarding the application of NMES, the limited effects on ventilator weaning were predictable since the investigators focused on the muscles of limbs rather than respiratory muscles [41, 42, 52, 53].” (Pages 17-18, Lines 296-320)

Methods

3. Please specify whether reports such as unpublished manuscripts or conference abstracts were eligible for inclusion

Reply: Thank you for your comments. The unpublished manuscripts or conference abstracts were not eligible for inclusion in this study. The selection criteria in the Study selection of Methods was more specified and revised as 

“The following inclusion criteria were used for study selection: (1) The target population was the critically ill patients with mechanical ventilation in the ICU rather than the chronic care center. (2) The interventions had to compare the control programs with lower intensity or frequency with experiment programs with higher dosage. (3) The outcome measures were focused on MV, such as ventilator duration or extubation rate. (4) Randomized controlled trials (RCT) in English or Chinese published in peer-reviewed journals and the studies provided information on the intervention protocol and dosage. Unpublished manuscripts and conference abstracts were not eligible for study selection. The exclusion criteria were studies without physiotherapy interventions or ventilator-related outcomes, and those focusing on interventions after extubation.” (Page 6, Lines 90-100)

4. Consider providing rationale for any notable restrictions to study eligibility (e.g., articles focusing on patients with chronic conditions.)

Reply: Thank you for your feedback. As you said, patients with chronic conditions may have difficulty weaning from ventilators too. However, our study focused on the patients with critical illness during the acute phase rather than the patients with specific diseases or chronic conditions. This is because being too weak to breath by themselves is a common situation among the patients in the ICUs and our study purpose is to investigate the solutions for them. Furthermore, having chronic conditions is one of the characteristics in the most ICU patients. The criteria of our study would include articles with various diagnoses and medical conditions, which covered the patients with comorbidities. According to the core concept of our research and the features of the target subjects, we further specified it as below:

“The following inclusion criteria were used for study selection: (1) The target population was the critically ill patients with mechanical ventilation in the ICU rather than the chronic care center.” (Page 6, Lines 90-92) 

5. How many and what were the Chinese languagues?

Reply: 

There was not any included article in our study published in Chinese. Although we searched randomized controlled trials in English or Chinese during the searching process, articles which met the criteria were all published in English.

6. Please consider specifying the date when each source was last searched or consulted. Specify restrictions on dates

Reply: 

Thank you for your comments. The manuscript was revised as 

“The concatenation of keywords and synonyms by “OR” and “AND” were searched in the following four databases on January 15, 2022: PubMed (1946–2021/12/31), Cochrane Library (1995–2021/12/31), EMBASE (1947–2021/12/31), and Airiti Library (1979–2021/12/31).” (Page 6-7, Lines 102-105)

7. Specify a bit more what “other sources” shown in S1 Appendix are

Reply: 

Thank you for your comments. The S1 Appendix was revised as 

“The search strategy for other sources (Hand searching) - Two reviewers identified additional references by checking the reference lists of identified articles.” (S1 Appendix)

8. Specify how articles were chosen from hand-searching references (what criteria were applied)?

Reply: 

Thank you for your comments. The criteria used in the database searching and hand searching was the same. The manuscript was revised as 

“In addition, handsearching was performed on the reference lists of included articles and previously published reviews” (Page 7, Lines 112-113)

9. Provide more details on search strategy as it was performed in each database.

Reply: 

Thank you for your comments. We added the detailed search strategy and search history of each database to the S1 Appendix. Please see the supporting material. 

As advised in the comment, the manuscript was revised as “The keywords included critical illness, intensive care unit, rehabilitation, physical therapy, early mobility, ventilator weaning, and extubation. Every synonym of the keywords was checked with MeSH and the same search protocol was used in each database. The detailed search strategy is shown in S1 Appendix.” (Page 7, Lines 105-109)

10. Did you need to use special tools to convert search terms between English and Chinese?

Reply: 

We did not use any tool to convert search terms between different languages. Airiti Library is an electronic database, which integrates academic resources from Taiwan and Mainland China. Its collected content covers important full-text content such as journal papers both in Chinese and English, and provides English Title, Abstract, and Keywords services for global scholars. Therefore, it is feasible to use English keywords to search related studies. That was why we did not need to convert search terms.

11. How were the articles translated?

Reply: 

We did not need to translate articles because the articles met the criteria in our study were published in the same language, English.

12. How was the data extracted (what software)?

Reply: 

Thank you for your comments. The manuscript was revised as 

“The data and results from the included studies were extracted by using a standardized spreadsheet of Excel (Microsoft Excel 2016) that documented basic information about the study” (Page 8, Lines 125-127)

13. Under Data Synthesis/Analysis, were there any limitations set to interpret the size of effect?

Reply: 

To interpret the size of effect, the manuscript was revised to clarify the criteria for meta-analysis and to minimize insufficient data for pooling as below.

“Meta-analysis was performed when the number of articles with the same ventilator-related outcome was more than three. The articles were excluded from pooling when there were data missing, insufficient treatment information, or difficulty in the grouping.” (under Data Synthesis/Analysis, Page 10, Lines 166-169)

We converted median to mean by the formulas from the previous literature which was published in BMC Medical Research Methodology.

“Outcomes reported as continuous variables were presented as the mean ± standard deviation. If only the median and interquartile range were reported, they were converted to mean and standard deviation using appropriate statistical formulas [30].” (under Data Extraction, Page 8, Lines 134-136) 

And we listed this possible bias as one of the study limitations. “(6) During the meta-analysis, there was one study that provided median rather than the mean data and we did not receive a response from the authors after contact via email [52]. In this study, we estimated the means based on formulas from the literature; however, some data extraction bias remains inevitable.” (Page 23, Lines 402-406)

14. What meta-analysis method was used?

Reply: 

The meta-analysis method was described in the part of Data Synthesis and Analysis. Here is the content in the manuscript. 

“The data synthesis used Review manager 5.4 with statistical significance set at a P-value of < 0.05. Categorical variables were compared using odds ratios (ORs), while continuous variables were compared using standardized mean difference (SMD). 95% confidence intervals (CIs) were calculated by Wald-type methods in Review manager 5.4 for all values. Heterogeneity among articles was assessed using the Q test and I2 statistic. When the I2 of overall heterogeneity was more than 50%, pooling data were analyzed in a random-effects model.” (Page 10, Lines 170-176)

15. Describe in detail process used to decide which studies of the 24 were eligible for synthesis (15?).

Reply: 

Because meta-analysis needed sufficient number of articles without missing data to constitute, we only chose the outcome which was provided in most articles to synthesis. Therefore, some articles using rare parameters as outcome measures, such as ventilator-free day, would be excluded from the meta-analysis. However, we still summarized these articles in the part of systematic review. 

As advised in the comment, the manuscript was revised as “Meta-analysis was performed when the number of articles with the same ventilator-related outcome was more than three. The articles were excluded from pooling when there were data missing, insufficient treatment information, or difficulty in the grouping. The detailed process of article selection for quantitative synthesis is shown in Fig 1.” (Page 10, Lines 166-170)

Finally, “Fifteen articles that used ventilator duration as an outcome measure met the criteria for meta-analysis. Regarding the types of treatment protocols, there were eight articles pooled for the meta-analysis, which used medical treatment or usual care as a comparison [31-33, 36, 37, 50-52].” (Page 14, Lines 235-238)

16. Please consider specifying method used to calculate the confidence interval for the summary effect (e.g., Wald-type CI, Hartung…..).

Reply:

Thank you for your comments. The manuscript was revised as “95% confidence intervals (CIs) were calculated by Wald-type methods in Review manager 5.4 for all values.” (Page 10, Lines 173-174)

Results

17. Please consider mentioning some studies that might have met inclusion criteria but were excluded from analyses and why?

Reply:

We excluded 9 articles for data synthesis. The reasons were shown in PRISMA flow diagram (Figure 1). More details were as the table in the file (Response to reviewers).

18. Please consider not starting consecutive sentences with “One study….”

Reply:

Thank you for your comments. The manuscript was revised as below: 

“One of the studies using early mobilization with/without an elastic band was beneficial with respect to ventilator duration when compared to multiple components including passive and active range of motion and breathing exercises [34].” (Page 13, Lines 210-212)

“A study using rotation therapy (changing position continuously for 18 hrs/day) and percussion showed significantly shorter ventilator duration and longer ventilator-free days than that of routine position changing every 2~4 hrs [36].” (Page 13, Lines 212-215)

“A study using multimodality chest physical therapy showed a higher extubation rate when compared to studies using manual hyperinflation and suctioning [37].” (Page 13, Lines 215-217)

Discussion

19. Please consider discussing limitations of your review process if any, and discuss impact of each limitation. You already discussed limitation of the evidence.

Reply:

Thank you for your comments. We further re-considered limitations of this review process and discuss impact of each limitation. The manuscript was revised as below. (Page 22-23, Lines 385-407)

“There were several limitations to this study: 

(1) there were 24 studies that could be included in the systematic review; however, only 15 studies were eligible to be included in the meta-analysis. After stratifying with type or frequency of intervention in comparison to medical treatment or low frequency, there were only fewer than five articles included in each meta-analysis. 

(2) Among the included articles with a low-to-moderate quality, the major risk of bias was a lack of blinding and placebo or true control. Thus, more high-quality randomized controlled trials with larger sample sizes in this field are essential. 

(3) Considering the diversity of ventilation-related outcome indicators and the small number of included studies, only the studies using the ventilation duration were included in the meta-analysis to ensure the reliability of the study and comparison. 

(4) Analyses of moderator variables on the effects of early rehabilitation programs (e.g, age, sex, sample size, comorbidities, sedation level) were not performed. Lesser clinical prognostic factors could be identified for the treatment plan. 

(5) Although the sedation level of subjects may affect their active participation in the intervention, only half of the included studies provided subjects’ conditions of sedation with various sedation controls among studies. Although we used the random-effects model for pooling, the high heterogeneity of sedation control might not be neglected and should be considered in clinical practice. 

(6) During the meta-analysis, there was one study that provided median rather than the mean data and we did not receive a response from the authors after contact via email [52]. In this study, we estimated the means based on formulas from the literature; however, some data extraction bias remains inevitable. 

(7) Publication bias was not suitable to be evaluated by funnel plot and the impact of bias may exist.”

20. Please consider providing recommendations for future research to address short comings of this study.

Reply:

Thank you for your comments. The manuscript was revised as 

” In the future, with the increase of research on the different protocols of early rehabilitation for patients in ICU, we plan to update this systematic review and meta-analysis. A network meta-analysis may be attempted to explore the association between the complexity and variety of multi-component protocols and outcomes.” (Page 23-24, Lines 412-416)

---

## [Decision Letter · Decision Letter 1]

13 Mar 2023

PONE-D-22-29326R1Effects of different types and frequencies of early rehabilitation on ventilator weaning among patients in intensive care units: A systematic review and meta-analysisPLOS ONE

Dear Dr. Tsai,

Thank you for submitting your manuscript to PLOS ONE. After careful consideration, we feel that it has merit but does not fully meet PLOS ONE’s publication criteria as it currently stands. Therefore, we invite you to submit a revised version of the manuscript that addresses the points raised during the review process. Please submit your revised manuscript by Apr 27 2023 11:59PM. If you will need more time than this to complete your revisions, please reply to this message or contact the journal office at plosone@plos.org. Please include the following items when submitting your revised manuscript:A rebuttal letter that responds to each point raised by the academic editor and reviewer(s). You should upload this letter as a separate file labeled 'Response to Reviewers'.A marked-up copy of your manuscript that highlights changes made to the original version. You should upload this as a separate file labeled 'Revised Manuscript with Track Changes'.An unmarked version of your revised paper without tracked changes. You should upload this as a separate file labeled 'Manuscript'.If applicable, we recommend that you deposit your laboratory protocols in protocols.io to enhance the reproducibility of your results. Protocols.io assigns your protocol its own identifier (DOI) so that it can be cited independently in the future. For instructions see: https://journals.plos.org/plosone/s/submission-guidelines#loc-laboratory-protocols. Additionally, PLOS ONE offers an option for publishing peer-reviewed Lab Protocol articles, which describe protocols hosted on protocols.io. Read more information on sharing protocols at https://plos.org/protocols?utm_medium=editorial-email&utm_source=authorletters&utm_campaign=protocols.

We look forward to receiving your revised manuscript.

Kind regards,

Steven E. Wolf, MD

Academic Editor

PLOS ONE

Journal Requirements:

Reviewers' comments:

Reviewer's Responses to Questions

**Comments to the Author**

1. If the authors have adequately addressed your comments raised in a previous round of review and you feel that this manuscript is now acceptable for publication, you may indicate that here to bypass the “Comments to the Author” section, enter your conflict of interest statement in the “Confidential to Editor” section, and submit your "Accept" recommendation.

Reviewer #1: (No Response)

Reviewer #2: All comments have been addressed

2. Is the manuscript technically sound, and do the data support the conclusions?

Reviewer #1: Yes

Reviewer #2: Yes

3. Has the statistical analysis been performed appropriately and rigorously? 

Reviewer #1: Yes

Reviewer #2: Yes

4. Have the authors made all data underlying the findings in their manuscript fully available?

Reviewer #1: Yes

Reviewer #2: Yes

5. Is the manuscript presented in an intelligible fashion and written in standard English?

Reviewer #1: Yes

Reviewer #2: Yes

6. Review Comments to the Author

Reviewer #1: (No Response)

Reviewer #2: Thank you for responding to all concerns and for a great job. just a few minor comments

Line 227, page 68 of the pdf document….Review Manager or manager? 5.4 capitalize the lettter "m" or not?

Line 271. Table 1. Europe, USA, South America are not Races or racial categories. They are geographical.

Lastly, I think that the native English speaker/writer did a wonderful job. However, there are still a few sections or segments or fragments that could use improvement.

7. PLOS authors have the option to publish the peer review history of their article (what does this mean?). If published, this will include your full peer review and any attached files.

Reviewer #1: **Yes: **Cassiano Teixeira

Reviewer #2: No

---

## [Author Response · Author response to Decision Letter 1]

23 Mar 2023

Journal Requirements:

Reply: 

We confirmed that our reference list was correct and complete without any retracted articles.

Reviewers' comments:

1. If the authors have adequately addressed your comments raised in a previous round of review and you feel that this manuscript is now acceptable for publication, you may indicate that here to bypass the “Comments to the Author” section, enter your conflict of interest statement in the “Confidential to Editor” section, and submit your "Accept" recommendation.

Reviewer #1: (No Response)

Reviewer #2: All comments have been addressed

Reply: Thank you for your thoughtful suggestions and insights, which have enriched the manuscript and produced a better and more balanced account of the research.

2. Is the manuscript technically sound, and do the data support the conclusions?

Reviewer #1: Yes

Reviewer #2: Yes

Reply: Thank you for your comment.

3. Has the statistical analysis been performed appropriately and rigorously?

Reviewer #1: Yes

Reviewer #2: Yes

Reply: Thank you for your comment.

4. Have the authors made all data underlying the findings in their manuscript fully available?

Reviewer #1: Yes

Reviewer #2: Yes

Reply: Thank you for your comment.

5. Is the manuscript presented in an intelligible fashion and written in standard English?

Reviewer #1: Yes

Reviewer #2: Yes

Reply: Thank you for your comment.

6. Review Comments to the Author

Reviewer #1: (No Response)

Reviewer #2: 

Thank you for responding to all concerns and for a great job. just a few minor comments

Line 227, page 68 of the pdf document….Review Manager or manager? 5.4 capitalize the letter "m" or not?

Line 271. Table 1. Europe, USA, South America are not Races or racial categories. They are geographical.

Lastly, I think that the native English speaker/writer did a wonderful job. However, there are still a few sections or segments or fragments that could use improvement.

Reply: 

Thanks for your comment. We revised Review manager to Review Manager and revised Race to Geographical region in the Table 1. In order to ensure the readability or logical flow of study, the manuscript was proofread and edited by native speaker again. All changes were highlighted in the file “Revised Manuscript with Track Changes”.

---

## [Editor Report · Decision Letter 2]

12 Apr 2023

Effects of different types and frequencies of early rehabilitation on ventilator weaning among patients in intensive care units: A systematic review and meta-analysis

PONE-D-22-29326R2

Dear Dr. Tsai,

We’re pleased to inform you that your manuscript has been judged scientifically suitable for publication and will be formally accepted for publication once it meets all outstanding technical requirements.

Kind regards,

Steven E. Wolf, MD

Academic Editor

PLOS ONE

---

## [Editor Report · Acceptance letter]

14 Apr 2023

PONE-D-22-29326R2 

Effects of different types and frequencies of early rehabilitation on ventilator weaning among patients in intensive care units: A systematic review and meta-analysis 

Dear Dr. Tsai:

I'm pleased to inform you that your manuscript has been deemed suitable for publication in PLOS ONE. Congratulations! Your manuscript is now with our production department. 

Kind regards, 

on behalf of

Dr. Steven E. Wolf 

Academic Editor

PLOS ONE